# Lifestyle Risk Factors and All-Cause and Cardiovascular Disease Mortality: Data from the Korean Longitudinal Study of Aging

**DOI:** 10.3390/ijerph16173040

**Published:** 2019-08-22

**Authors:** Inhwan Lee, Shinuk Kim, Hyunsik Kang

**Affiliations:** 1College of Sport Science, Sungkyunkwan University, Suwon 16419, Korea; 2Department of Civil Engineering, Sangmyung University, Cheonan 31066, Korea

**Keywords:** unhealthy behaviors, relative risk, premature death, Korean adults

## Abstract

This study examined the association between lifestyle risk factors and all-cause and cardiovascular disease (CVD) mortality in 9945 Korea adults (56% women) aged 45 years and older. Smoking, heavy alcohol intake, underweight or obesity, physical inactivity, and unintentional weight loss (UWL) were included as risk factors. During 9.6 ± 2.0 years of follow-up, there were a total of 1530 cases of death from all causes, of which 365 cases were from CVD. Compared to a zero risk factor (hazard ratio, HR = 1), the crude HR of all-cause mortality was 1.864 (95% CI, 1.509–2.303) for one risk factor, 2.487 (95% confidence interval, CI, 2.013–3.072) for two risk factors, and 3.524 (95% CI, 2.803–4.432) for three or more risk factors. Compared to a zero risk factor (HR = 1), the crude HR of CVD mortality was 2.566 (95% CI, 1.550–4.250) for one risk factor, 3.655 (95% CI, 2.211–6.043) for two risk factor, and 5.416 (95% CI, 3.185–9.208) for three or more risk factors. The HRs for all-cause and CVD mortality remained significant even after adjustments for measured covariates. The current findings showed that five lifestyle risk factors, including smoking, at-risk alcohol consumption, underweight/obesity, physical inactivity, and UWL, were significantly associated with an increased risk of all-cause and CVD mortality in Korean adults.

## 1. Introduction

A large proportion of global deaths from all causes and diseases has been attributed to modifiable risk factors, including smoking, heavy alcohol intake, unhealthy body weights, physical inactivity, and unhealthy diets (https://www.who.int/gho/ncd/risk_factors/en/). On the other hand, healthy behaviors, including non-smoking, moderate alcohol intake, healthy body weights, physical activity, and healthy diets, are associated with a lower risk of premature death from all causes and diseases [1].

Along with those existing risk factors, unintentional weight loss (UWL) is also emerging as an independent risk factor. Findings from previous studies showed that UWL was significantly associated with a higher risk of premature death from all causes and cardiovascular disease (CVD), especially in geriatric populations [2,3]. Thus, adding UWL to the existing risk factors will provide more meaningful information for better understanding the etiology of premature death from all causes and CVD. 

Similarly to in Western populations, lifestyle risk factors are also known to be significant determinants of health conditions in Korean populations [4,5], especially among individuals with low-socioeconomic status (i.e., low income, less educated, and living alone) [6]. Lifestyle risk factors are associated with the prevalence of prostate cancer and breast cancer in Korean men and women, respectively [7]. On the other hand, healthy lifestyle behaviors characterized by regular diet, exercise, and health screening, are associated with better physical and mental health conditions in Korean adults [8]. 

Participants in the Korean longitudinal study of aging (KLoSA) have his/her unique lifestyle pattern characterized by higher rates of smoking, heavy alcohol intake, and physical inactivity [9]. Despite the high prevalence of lifestyle risk factors in Korea [10], however, only one population-based study has been conducted to examine the association between lifestyle risk factors and mortality, reporting a significant relationship of unhealthy behaviors with an increased risk of all-cause, cancer, and non-cancer mortality in Korean adults [11]. More studies at a population level are necessary to provide more useful and practical information for policy making and diseases prevention in Korea. 

Furthermore, the association between lifestyle risk factors and premature deaths from all causes and CVD may be modulated by additional covariates, including ethnicity, culture, religion, employment status, region, and others. Thus, exploring the associations between lifestyle risk factors and all-cause and CVD mortality in the KLoSA population would be of great significance for men and women in other cities of Korea and for men and women in other Asian countries with similar culture and lifestyle. In this population-based study, therefore, we investigated the association between lifestyle risk factors and all-cause and CVD mortality in Korean adults.

## 2. Materials and Methods

### 2.1. Study Design and Participants

The KLoSA is a large population-based survey study conducted nationwide in the Republic of Korea that launched in 2006. A detailed description of the KLoSA is provided elsewhere [11,12]. In brief, the KLoSA survey included Korean adults are who aged 45 years or older living in 15 metropolitan cities and provinces of South Korea. The KLoSA survey was conducted by using a multi-stage, stratified sampling based on the geographical areas and housing types across the nation. A total of 10,254 persons were included as panel respondents (1.7 persons per household) from a total of 6171 households, with a 70.7% household response rate and a 75.4% individual response rate within households. 

The KLoSA protocol was reviewed and approved by the Institutional Review Board of Statistics Korea (approval number: 336052). All of the participants provided written informed consent. Trained surveyors collected data by conducting computer-assisted face-to-face interviews. Among the total 10,254 samples, people who have no data on education (*n* = 7), body mass index (BMI) (*n* = 241), drinking (*n* = 59), and smoking (*n* = 2) were excluded. After the selection process, the final analytical sample (*n* = 9945) was restricted to 4345 men and 5600 women. KLoSA is accessible via the National public database (http://www.kli.re.kr/klosa/en/about/introduce.jsp).

### 2.2. Study Variables

#### 2.2.1. Lifestyle Risk Factors

Lifestyle risk factors included in the current study were smoking (currently or past smoker < 10 years ago), at-risk alcohol consumption (≥5 drinks per week) [13], underweight (<18.5) or obesity (BMI ≥ 25), physical inactivity (less than 2 times per week), or UWL (weight loss of 5 kg or more). For each risk factor, a binary score (yes = 1 or no = 0) was given. Consequently, risk score ranged from 0 (healthiest) to 5 (least healthy) by summing the binary scores for each of the five risk factors. For the purposes of data analysis, we merged the 3, 4, and 5 risk scores into ≥3 risk scores due to the small number of exposed risk factors.

#### 2.2.2. Covariates

Covariates used in this study include age, marital status (married, widow/divorced, unmarried), educational level (elementary school or less, middle school, high school, and college or more), household income, employment status (currently employed or not employed), religion (no religion or protestant, catholic, Buddhist, and others), type of housing (apartment or general house), and region (urban or rural). In addition, the Mini-mental Status Examination (K-MMSE), Activities of Daily Living (K-ADL), and Center for Epidemiological Studies of Depression 10 (CES-D 10) were used to assess cognitive function, ADL, and depressive symptoms, respectively. Self-reported disease histories were based on physician-diagnosed diseases, including hypertension, diabetes mellitus, cardiac disorders, gastrointestinal diseases, arthritis, cancer, lung disease, and stroke [11,12].

#### 2.2.3. Outcomes

The primary outcomes of the study were CVD- and non-CVD-mortality due to not enough cases being available to analyze statistically, as well as all-cause mortality. We confirmed mortality outcomes by cross-checking the acquired information with the death records from the national statistical office in Korea. 

### 2.3. Statistical Analyses

Multiple imputation was used to handle the missing values of the major study variables identified (<8%) prior to the primary analyses. The number of imputations was determined to be four. Baseline characteristics were presented as mean ± standard deviation (SD) or percentage. Characteristics were compared between men and women and according to number of lifestyle risk factors using Pearson’s χ^2^ test for categorical variables and analysis of variance for continuous variables. The Kaplan-Meier procedure with log-rank tests was used to estimate mortality functions according to a number of baseline lifestyle risk factors. Survival time was measured as the time from the baseline survey to death or the censor point (30 November 2016). The Cox proportional hazards models were used to estimate hazard ratios (HRs) and 95% confidence intervals (CIs) for all-cause and CVD- and non-CVD-mortality. All analyses were carried out taking into account complex sampling weights, using SPSS-PC 23.0 (SPSS Inc., Chicago, IL, USA).

## 3. Results

The final sample for analysis included 9945 participants and had an average follow-up of 9.6 ± 2.0 years for a total of 95,632 person-years. During the follow-up period, there were a total of 1530 cases of death from all causes, of which 365 cases were from CVD. As illustrated in Table 1, the mean age at study entry was 61.5 ± 11.0 years for all study participants, 61.1 ± 10.5 years for men and 61.8 ± 11.3 years for women. Women had lower household income, lower marital status, less education, and lower employment rates than men. In addition, men had higher rates of cognitive impairment, depressive symptoms, comorbidity, medications, and falls than women. With respect to lifestyle risk factors, men were more active but had higher rates of at-risk alcohol consumption and smoking than women. No significant differences in BMI, ADL impairment, hospitalization, UWL, religion, type of housing, and region were found between women and men.

Covariates included in this study are presented according to increasing number of lifestyle risk factors, as illustrated in Table 2. In general, significant linear trends in mean age, BMI, education, occupation, type of housing, religion, region, cognitive impairment, depressive symptoms, and ADL impairment were found according to an increasing number of lifestyle risk factors; individuals with one or more lifestyle risk factors were likely to be older, heavier, less educated, unemployed, have an absence of religion, reside in a general house, and live in a rural area, but had a lower household income and higher rates of cognitive impairment, depressive symptoms, and ADL impairments, compared to individuals with zero risk factor. No significant linear trends were found in marital status, comorbidity, medications, falls, and hospitalization according to a number of lifestyle risk factors.

As illustrated in Table 3, the prevalence of lifestyle risk factors was 29.1% (95% CI, 28.2–30.0%) for smoking, 5.9% for at-risk alcohol consumption (95% CI, 5.5–6.4%), 26.3% (95% CI, 25.5–27.2%) for underweight/obesity, 71.3% (95% CI, 70.4–72.2%) for physical inactivity, and 10.3% (95% CI, 9.8–11.0%) for UWL. Except for underweight/obesity, all risk factors, including smoking, at-risk alcohol consumption, physical inactivity, and UWL, were individually associated with increased risks of premature deaths from all causes and CVD.

The relative risks of deaths from all causes and CVD are presented according to increasing number of the 4 or 5 risk factors, so as to examine the additive impact of UWL on mortality, as illustrated in Table 4 and Table 5. 

When 4 risk factors-smoking, at-risk alcohol consumption, physical inactivity, and underweight/obesity-were considered, the crude HR for all-cause mortality was 1.747 (95% CI, 1.445–2.111) for one risk factor, 2.194 (95% CI, 1.809–2.661) for two risk factors, and 2.699 (95% CI, 2.148–3.392) for three or more risk factors, compared to zero risk factor (HR = 1). The HRs for two and three or more risk factors remained significant (P = 0.029 and P < 0.001, respectively) even after adjustments for all the covariates, while the HR for one risk factor was no longer significant (P = 0.154) after adjustments for all the covariates. The crude HR for CVD mortality were 1.841 (95% CI, 1.223–2.772) for one risk factor, 2.586 (95% CI, 1.711–3.907) for two risk factors, and 3.474 (95% CI, 2.168–5.567) for three or more risk factors, as compared to zero risk factor (HR = 1). The HR for three or more risk factors remained significant (P = 0.010) even after adjustments for all the covariates, while the HRs for one and two risk factors were no longer significant (P = 0.500 and P = 0.139, respectively) after adjustments for all the covariates. Finally, the crude HR for non-CVD mortality was 1.721 (95% CI, 1.390–2.131) for one risk factor, 2.088 (95% CI, 1.678–2.599) for two risk factors, and 2.490 (95% CI, 1.916–3.236) for three or more risk factors, as compared to zero risk factor (HR = 1). The HR for three or more risk factors remained significant (P = 0.006) even after adjustments for all the covariates, while the HRs for one and two risk factors were no longer significant (P = 0.207 and P = 0.096, respectively) after adjustments for all the covariates.

When all 5 risk factors—4 risk factors plus UWL—were considered, the crude HR for all-cause mortality was 1.864 (95% CI, 1.509–2.303) for one risk factor, 2.487 (95% CI, 2.013–3.072) for two risk factors, and 3.524 (95% CI, 2.803–4.432) for three or more risk factors, compared to zero risk factor (HR = 1). The HRs for one (P = 0.022), two (P = 0.006), and three or more risk factors (P < 0.001) remained statistically significant even after adjustments for all the covariates. The crude HR for CVD mortality was 2.566 (95% CI, 1.550–4.250) for one risk factor, 3.655 (95% CI, 2.211–6.043) for two risk factors, and 5.416 (95% CI, 3.185–9.208) for three or more risk factors, compared to zero risk factor (HR = 1). The HRs for one (P = 0.036), two (P = 0.012), and three or more risk factors (P = 0.001) remained statistically significant, even after adjustments for all the covariates. Finally, the crude HR for non-CVD mortality was 1.722 (95% CI, 1.364–2.174) for one risk factor, 2.250 (95% CI, 1.782–2.842) for two risk factors, and 3.141 (95% CI, 2.433–4.055) for three or more risk factors. The HRs for two and three or more risk factors remained significant (P = 0.047 and P < 0.001, respectively) even after adjustments for all the covariates, while the HR for one risk factor was no longer significant (P = 0.115) after adjustments for all the covariates.

As illustrated in Figure 1, the Kaplan-Meier mortality functions showed that the relative risks of all-cause and CVD- and non-CVD mortality increased significantly according to an increasing number of lifestyle risk factors.

## 4. Discussion

In this population-based prospective study, we examined the associations between lifestyle risk factors and all-cause and CVD mortality during 10 years of follow-up in Korean adults and found that individual and combined lifestyle risk factors—smoking, at-risk alcohol consumption, underweight/obesity, physical inactivity, and UWL—were significantly associated with an increased risk of all-cause and CVD- and non-CVD mortality independent of potential covariates. 

The current findings support and extend those of previous studies reporting the associations between health behaviors and premature death from all and specific causes in Western populations; the healthier the behaviors, the lower the mortality risk or vice versa. By analyzing data from a healthy ageing: a longitudinal study in Europe (HALE) study, for example, Knoops et al. [14] showed that adherence to healthy lifestyle behaviors, including the Mediterranean diet, physical activity, moderate alcohol consumption, and not smoking, was associated with a lower rate of all-cause and specific-cause mortality in European older adults. Using data obtained from a nurses’ health study (n = 78,865) and a health professionals follow-up study (n = 44,354), Li et al. [15] also examined five low-risk lifestyle factors (i.e., non-smoking, healthy body weights, physical activity, moderate alcohol intake, and a healthy diet) and mortality and found that compared with those who had zero-low risk factor (HR, 1), those who had 5 low-risk factors had significantly lower risks of all-cause mortality (HR, 0.26; 95% CI, 0.22–0.31), cancer mortality (HR, 0.35; 95% CI, 0.27–0.45), and CVD mortality (HR, 0.18; 95% CI, 0.12–0.26). 

On the other hand, Krokstad et al. [16] examined the association between lifestyle risk factors—including smoking, alcohol consumption, diet, physical activity, sedentary behaviors, sleep, and social participation—and mortality in a Norwegian cohort study (n = 36,911 adults aged 20~69 years) and found that all the risk factors, except for diet, were significantly associated with an increased risk of 14.1-year-all-cause and CVD mortality. By conducting a population-based cohort study (n = 231,048 Australians aged 45 years or older), Ding et al. [17] also examined the associations between six risk factors—smoking, alcohol use, dietary behavior, physical inactivity, sedentary behavior, and sleep—and mortality and found a positive linear trend of mortality risk according to an increasing number of risk factors. In particular, combinations of physical inactivity, prolonged sitting, and long sleep duration as well as combinations of smoking and high alcohol use were the most important determinants of all-cause mortality, implying individual as well as additive effects of those lifestyle risk factors on mortality. Together, those findings including the current ones support the associations between lifestyle behaviors and the risk of premature death from all causes and diseases in Asian as well as Western populations.

Similarly to in Western populations, lifestyle risk factors are also associated with risks of morbidities, including hypertension [18], prostate cancer, breast cancer, and others in Korea [7]. By analyzing data from the 5th Korean National Health and Nutrition Examination Survey dataset 2010–2012, Ha et al. [6] found that unhealthy behaviors, including smoking, heavy drinking, poor diet, and physical inactivity, had a higher tendency to cluster among men who were younger, had less education, and were living alone. 

In addition to morbidity, lifestyle risk factors are also associated with the risk of mortality in Korea. For example, Yun et al. [11] found that unhealthy behaviors, including smoking, heavy alcohol intake, overweight or obese weight, physical inactivity, and unhealthy diet, were associated with increased 10.2-year death risks from all causes, cancer, and non-cancer among middle-aged and older Korean adults. By analyzing data from the Seoul male cohort study that included 12,538 middle-aged and older men, Kim et al. [19] found that cardiovascular health behaviors—smoking, high blood pressure, and high fasting blood glucose—were associated with an increased 19-year risk of all-cause and CVD mortality. The adjusted population attributable risks of the 3 combined risk factors for all-cause and CVD mortality were 35.2% (95% CI, 21.7–47.4) and 52.8% (95% CI, 22.0–74.0), respectively. 

In addition to 4 existing risk factors, such as smoking, at-risk alcohol consumption, underweight/obesity, and physical inactivity, we are the first to report that adding UWL as a new risk factor magnified the impact of unhealthy behaviors in relation to all-cause and CVD mortality in Korea adults. In support of the current finding, De Stefani et al. [2] examined the association between UWL and all-cause and CVD mortality by conducting a meta-analysis of 15 studies that included a total of 178,644 participants and found that adjusted risk ratios of all-cause mortality and major cardiovascular events were 1.38 (95% CI, 1.23–1.53) and 1.17 (95% CI, 0.98–1.37), respectively. 

With respect to weight loss as a risk factor, intentionality appears to be a key factor in determining the nature of its impact on mortality. In a meta-analysis of 15 clinical studies that included 17,186 participants (53% women), for example, Kritchevsky et al. [20] found that individuals with 5.5 of weight loss had a 15% lower all-cause mortality risk (risk ratio, RR, 0.85; 95% CI, 0.73–1.00). Similarly, Lee et al. [21] examined the association between weight loss and all-cause mortality in Korean elderly persons and found that UWL was associated with an increased risk of all-cause mortality (RR, 1.82; 95% CI, 1.58–2.08), whereas intentional weight loss was associated with an decreased risk all-cause mortality (RR, 0.61; 95% CI, 0.43–0.85).

Several explanations can be given for the individual and combined impacts of lifestyle risk factors on all-cause and CVD mortality found in the current study. First, tobacco smoking is a well-established and leading cause of premature death worldwide [22]. A meta-analysis of previous studies investigating the association between smoking and mortality in older adults showed that compared with never-smokers, current and former smokers had higher risks of all-cause mortality of 83% and 34%, respectively [23]. 

Second, heavy alcohol consumption has been found to be associated with a higher risk of chronic diseases such CVD and dementia, thereby increasing risk of premature death from the diseases [24]. 

Third, obesity/overweight is an established risk factor for causes of premature deaths from CVD and cancers [25]. In addition, being underweight is associated with existing illness or health conditions, thereby leading to an increased risk of premature deaths from diseases. With respect to unhealthy body weights, smoking may play as an important confounder that should be taken into account since smokers tend to have lower body weights and higher mortality than nonsmokers [26].

Fourth, UWL is also associated with existing illness or health conditions, especially in older persons [27], and thus is also associated with mortality [28]. 

Lastly, physical activity is associated with a lower risk of all-cause and cause-specific mortality. Like in young adults, it is well known that older adults may have numerous health benefits from participating in physical activity of any intensity, including improvements of cardiovascular, metabolic, and immune system, collectively contributing the ability to maintain homeostasis. In particular, regular physical activity may prevent and/or ameliorate aging-associated decline in the β adrenergic responsiveness indicative of the overall function of the cardiovascular system, contributing to the clinical improvement in the cardiovascular system and decreased risks of CVD morbidity and mortality in elderly persons [29]. In addition, Paganini-Hill et al. [30] by conducting a study of approximately 13,000 elderly people showed that regardless of time spent in physical activity, active individuals had a 15–35% lower risk of mortality during 28 years of follow-up than inactive individuals. In another population-based cohort study involving 2357 men aged ≥ 65 years, Yates et al. [31] showed that compared with individuals who exercised 1 or fewer times per week, individuals who exercised vigorously 2–4 times per week had a lower risk of 28-year-mortality by 15–35%. Together, the current findings show that having unhealthy behaviors contribute to an increased risk of all-cause and CVD mortality individually as well as additively in Korean adults.

The present study has some limitations. First, the cross-sectional nature of the current study may limit its ability to test the impacts of lifestyle risk factors in relation to all-cause and CVD-and non-CVD mortality in a cause-and-effect manner. Therefore, the current findings should be confirmed via a longitudinal study to establish causal relationships between unhealthy behaviors and premature deaths from all causes and CVD and non-CVD in Korea adults. Second, the current study did not include other significant risk factors, such as poor nutrition, prolonged sitting, and sleeping, which may affect health conditions and mortality. In particular, poor quality of diets (e.g., high-carbohydrate or high-fat) may contribute to an increased risk of CVD morbidity [32] and mortality [33] by triggering oxidative stress and thereby inflammatory responses [34]. Unfortunately, however, the KloSA data failed to include dietary information from the participants. Future studies should include those risk factors to delineate the complexity of the association between lifestyle risk factors and mortality in a better way.

Despite the limitations, the current study has shed light on the relationships between lifestyle risk factors and all-cause and CVD- and non-CVD mortality in Korean adults. To the best of our knowledge, no study has examined whether adding UWL to existing risk factors strengthens the power to elucidate the association between the risk factors and all-cause and CVD and non-CVD mortality in Korean adults. In this respect, the current findings contribute to the literature on unhealthy behaviors in relation to all-cause and CVD- and non-CVD mortality in Asian countries.

## 5. Conclusions

In the current study, we examined the association between lifestyle risk factors and mortality from all causes and CVD in a representative sample of Korean adults and found that all five risk factors, including smoking, at-risk alcohol consumption, underweight or obesity, physical inactivity, and UWL, were significantly associated with an increased risk of all-cause and CVD- and non-CVD mortality, implying the urgency of modifying unhealthy lifestyle behaviors to ameliorate disease burden and mortality in Korean adults.

## Figures and Tables

**Figure 1 ijerph-16-03040-f001:**
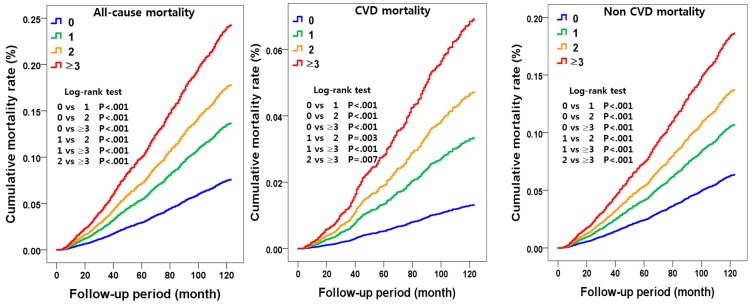
The Kaplan-Meier survival curves for all-cause and CVD and non CVD mortality according to a number of lifestyle risk factors.

**Table 1 ijerph-16-03040-t001:** Descriptive statistics of study participants.

Variables	Total (*n* = 9945)	Men (*n* = 4345)	Women (*n* = 5600)	P Value
Age (years)	61.5 ± 11.0	61.1 ± 10.5	61.8 ± 11.3	0.002
BMI (kg/m^2^)	23.3 ± 3.3	23.2 ± 2.8	23.3 ± 3.6	0.082
Household income (10,000 won)	159.9 ± 208.2	173.6 ± 208.4	149.0 ± 207.4	<0.001
Marital status, (n %)				<0.001
Married	7769 (78.1)	3997 (92.0)	3772 (67.4)	
Widow/divorced	2095 (21.1)	303 (7.0)	1792 (32.0)	
Unmarried	81 (0.8)	45 (1.0)	36 (0.6)	
Education, n (%)				<0.001
Elementary or less	4644 (46.7)	1375 (31.6)	3269 (58.4)	
Middle and high school	4273 (43.0)	2211 (50.9)	2062 (36.8)	
College or higher	1028 (10.3)	759 (17.5)	269 (4.8)	
Employment status, n (%)				<0.001
Yes	3889 (39.1)	2510 (57.8)	1379 (24.6)	
No	6056 (60.9)	1835 (42.2)	4221 (75.4)	
Religion, n (%)				<0.001
No religion	4424 (44.5)	2405 (55.4)	2019 (36.1)	
Protestant	1990 (20.0)	719 (16.5)	1271 (22.7)	
Catholic	897 (9.0)	309 (7.1)	588 (10.5)	
Buddhist	2530 (25.5)	875 (20.1)	1655 (29.5)	
Others	104 (1.0)	37 (0.9)	67 (1.2)	
Type of housing, n (%)				0.617
Apartment	3676 (37.0)	1618 (37.2)	2058 (36.8)	
General house	6269 (63.0)	2727 (62.8)	3542 (63.2)	
Region, n (%)				0.997
Urban	7688 (77.3)	3359 (77.3)	4329 (77.3)	
Rural	2257 (22.7)	986 (22.7)	1271 (22.7)	
Impaired cognition, n (%)	2692 (27.1)	747 (17.2)	1945 (34.7)	<0.001
Depressive symptoms, n (%)	3144 (31.6)	1083 (24.9)	2061 (36.8)	<0.001
Impaired ADL, n (%)	230 (3.9)	159 (3.7)	230 (4.1)	0.253
Falls, n (%)	396 (4.0)	92 (2.1)	304 (5.4)	<0.001
Hospitalization, n (%)	1105 (11.1)	463 (10.7)	642 (11.5)	0.203
Comorbidity, n (%)				<0.001
0	5266 (53.0)	2524 (58.1)	2742 (49.0)	
1	2863 (28.8)	1186 (27.3)	1677 (29.9)	
≥2	1816 (18.3)	635 (14.6)	1181 (21.1)	
Medications, n (%)				<0.001
0	5997 (60.3)	2809 (64.6)	3188 (56.9)	
1	2581 (26.0)	1062 (24.4)	1519 (27.1)	
≥2	1367 (13.7)	474 (10.9)	893 (15.9)	
Lifestyle risk factors				
Past/current smoking, n (%)	2892 (29.1)	2671 (61.5)	221 (3.9)	<0.001
At-risk alcohol consumption, n (%)	590 (5.9)	541 (12.5)	49 (0.9)	<0.001
Physical inactivity, n (%)	7095 (71.3)	2995 (68.9)	4100 (73.2)	<0.001
Underweight/obesity, n (%)	2617 (26.3)	1073 (24.7)	1544 (27.6)	0.001
Weight loss, n (%)	1029 (10.3)	427 (9.8)	602 (10.8)	0.134

BMI: body mass index, ADL: activities of daily living.

**Table 2 ijerph-16-03040-t002:** Descriptive statistics of measured parameters according to lifestyle risk factors.

	Number of Lifestyle Risk Factors				P for Linear Trend
Variables	0 (13.3%)	1 (42.9%)	2 (32.8%)	≥3 (11.0%)	
Women, n (%)	946 (71.6)	2996 (70.3)	1462 (44.8)	196 (17.9)	<0.001
Age (years)	59.3 ± 10.1	61.6 ± 11.0	62.2 ± 11.0	61.8 ± 11.2	<0.001
BMI (kg/m^2^)	22.6 ± 1.5	22.8 ± 2.4	23.9 ± 4.0	23.9 ± 4.6	<0.001
Household income (10,000 won)	203.4 ± 267.0	157.6 ± 222.3	149.0 ± 172.2	149.2 ± 156.8	<0.001
Marital status (n %)					0.353
Married	1098 (83.1)	3199 (75.0)	2558 (78.4)	914 (83.3)	
Widow/divorced	219 (16.6)	1036 (24.3)	669 (20.5)	171 (15.6)	
Unmarried	4 (0.3)	29 (0.7)	36 (1.1)	12 (1.1)	
Education, n (%)					<0.001
Elementary or less	430 (32.6)	2112 (49.5)	1594 (48.9)	508 (46.3)	
Middle or high school	658 (49.8)	1759 (41.3)	1382 (42.4)	474 (43.2)	
College or greater	233 (17.6)	393 (9.2)	287 (8.8)	115 (10.5)	
Occupation, n (%)					<0.001
Yes	388 (29.4)	1501 (35.2)	1409 (43.2)	591 (53.9)	
No	933 (70.6)	2763 (64.8)	1854 (56.8)	506 (46.1)	
Religion, n (%)					<0.001
No religion	435 (32.9)	1748 (41.0)	1611 (49.4)	630 (57.4)	
Protestant	315 (23.8)	976 (22.8)	568 (17.4)	131 (11.9)	
Catholic	162 (12.3)	378 (8.9)	284 (8.7)	73 (6.7)	
Buddhist	393 (29.8)	1108 (26.0)	773 (23.7)	256 (23.4)	
Other	16 (1.2)	54 (1.3)	27 (0.8)	7 (0.6)	
Type of housing, n (%)					<0.001
Apartment	679 (51.4)	1572 (36.9)	1110 (34.0)	315 (28.7)	
General house	642 (48.6)	2692 (63.1)	2153 (66.0)	782 (71.3)	
Region, n (%)					<0.001
Urban	1157 (87.6)	3282 (77.0)	2449 (75.1)	800 (72.9)	
Rural	164 (12.4)	982 (23.0)	814 (24.9)	297 (27.1)	
Impaired cognition, n (%)	215 (16.3)	1256 (29.5)	938 (28.7)	283 (25.8)	<0.001
Depressive symptom, n (%)	307 (23.2)	1344 (31.5)	1104 (33.8)	389 (35.5)	<0.001
Impaired ADL, n (%)	22 (1.7)	146 (3.4)	153 (4.7)	68 (6.2)	<0.001
Falls, n (%)	42 (3.2)	184 (4.3)	124 (3.8)	46 (4.2)	0.275
Hospitalization, n (%)	121 (9.2)	447 (10.5)	397 (12.2)	140 (12.8)	<0.001
Co-morbidity, n (%)					<0.001
0	784 (59.4)	2298 (53.9)	1626 (49.8)	558 (50.9)	
1	349 (26.4)	1206 (28.3)	979 (30.0)	329 (30.0)	
≥2	188 (14.2)	760 (17.8)	658 (20.2)	210 (19.1)	
Medications, n (%)					<0.001
0	870 (65.9)	2603 (61.0)	1868 (57.2)	656 (59.8)	
1	317 (24.0)	1099 (25.8)	876 (26.8)	289 (26.3)	
≥2	134 (10.1)	562 (13.2)	519 (15.9)	152 (13.9)	

BMI: body mass index, ADL: activities of daily living.

**Table 3 ijerph-16-03040-t003:** Prevalence of lifestyle risk factors at baseline and hazard ratios of all-cause and CVD mortality during 10 years of follow-up.

Lifestyle Risk Factors	Prevalence (%) at Baseline (95% CI)	Death Risk from All Causes, HR (95% CI)	Death risk from CVD, HR (95% CI)	Death Risk from non CVD, HR (95% CI)
Current/past smoking	29.1% (28.2–30.0)	1.603 (1.447–1.777)	1.857 (1.509–2.286)	1.530 (1.359–1.722)
At-risk alcohol consumption	5.9% (5.5–6.4)	1.473 (1.228–1.767)	1.669 (1.173–2.376)	1.413 (1.143–1.746)
Physical inactivity	71.3% (70.4–72.2)	1.479 (1.310–1.669)	1.635 (1.267–2.110)	1.435 (1.250–1.646)
Underweight/obesity	26.3% (25.5–27.2)	0.991 (0.884–1.110)	1.013 (0.803–1.278)	0.984 (0.863–1.121)
Unintentional weight loss	10.3% (9.8–11.0)	2.001 (1.755–2.283)	2.029 (1.551–2.653)	1.993 (1.713–2.318)

HR: hazard ratio, CI: confidence interval, CVD: cardiovascular disease.

**Table 4 ijerph-16-03040-t004:** The risks of all cause- and cardiovascular disease-mortality according to a number of lifestyle risk factors.

	Number of Lifestyle Risk Factors, Including Smoking, Heavy Alcohol Use, Physical Inactivity, and Underweight/Obesity						
	0	1		2		≥3	
All-cause mortality		HR (95% CI)	P value	HR (95% CI)	P value	HR (95% CI)	P value
Death (n, %)	127 (8.7)	682 (14.6)		546 (18.0)		175 (21.8)	
Crude HR (95% CI)	1	1.747 (1.445–2.111)	<0.001	2.194 (1.809–2.661)	<0.001	2.699 (2.148–3.392)	<0.001
Adjusted HR (95% CI)	1	1.155 (0.947–1.410)	0.154	1.259 (1.024–1.546)	0.029	1.609 (1.252–2.068)	<0.001
CVD mortality							
Death (n, %)	27 (1.9)	153 (3.3)		137 (4.5)		48 (6.0)	
Crude HR (95% CI)	1	1.841 (1.223–2.772)	0.003	2.586 (1.711–3.907)	<0.001	3.474 (2.168–5.567)	<0.001
Adjusted HR (95% CI)	1	1.155 (0.760–1.756)	0.500	1.382 (0.900–2.122)	0.139	1.940 (1.171–3.214)	0.010
Non-CVD mortality							
Death (n, %)	100 (6.9)	529 (11.4)		409 (13.5)		127 (15.8)	
Crude HR (95% CI)	1	1.721 (1.390–2.131)	<0.001	2.088 (1.678–2.599)	<0.001	2.490 (1.916–3.236)	<0.001
Adjusted HR (95% CI)	1	1.156 (0.923–1.449)	0.207	1.211 (0.965–1.544)	0.096	1.508 (1.128–2.015)	0.006

Adjusted for age, sex, household income, marital status, education, occupation, religion, type of housing, region, cognition, depressive symptoms, comorbidity, medications, ADL condition, fall, and hospitalization. HR: hazard ratio, CI: confidence interval, CVD: cardiovascular disease, ADL: activities of daily living.

**Table 5 ijerph-16-03040-t005:** Risk of mortality during 10 years of follow-up according to a number of lifestyle risk factors at the baseline.

	Lifestyle Risk Factors: Physical Inactivity, Heavy Alcohol Intake, Smoking, Underweight/Obesity, Unintentional Weight Loss						
	0	1		2		≥ 3	
All-cause mortality		HR (95% CI)	P value	HR (95% CI)	P value	HR (95% CI)	P value
Death (n, %)	101 (7.6)	582 (13.6)		581 (17.8)		266 (24.2)	
Crude HR (95% CI)	1	1.864 (1.509–2.303)	<0.001	2.487 (2.013–3.072)	<0.001	3.524 (2.803–4.432)	<0.001
Adjusted HR (95% CI)	1	1.302 (1.043–1.625)	0.020	1.396 (1.116–1.748)	0.004	1.871 (1.457–2.401)	<0.001
CVD mortality							
Death (n, %)	17 (1.3)	135 (3.2)		144 (4.4)		69 (6.3)	
Crude HR (95% CI)	1	2.566 (1.550–4.250)	<0.001	3.655 (2.211–6.043)	<0.001	5.416 (3.185–9.208)	<0.001
Adjusted HR (95% CI)	1	1.682 (1.008–2.805)	0.046	1.885 (1.127–3.152)	0.016	2.595 (1.487–4.526)	0.001
Non-CVD mortality							
Death (n, %)	84 (6.4)	447 (10.5)		437 (13.4)		197 (18.0)	
Crude HR (95% CI)	1	1.722 (1.364–2.174)	<0.001	2.250 (1.782–2.842)	<0.001	3.141 (2.433–4.055)	<0.001
Adjusted HR (95% CI)	1	1.219 (0.953–1.560)	0.115	1.288 (1.003–1.654)	0.047	1.710 (1.291–2.265)	<0.001

Adjusted for age, sex, household income, marital status, education, occupation, religion, type of housing, region, cognition, depressive symptoms, comorbidity, medications, ADL condition, fall, and hospitalization. HR: hazard ratio, CI: confidence interval, CVD: cardiovascular disease, ADL: activities of daily living.

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
