# Peer review of "Lifestyle Risk Factors and All-Cause and Cardiovascular Disease Mortality: Data from the Korean Longitudinal Study of Aging"

_ijerph, 2019, doi:10.3390/ijerph16173040_

Round 1

Reviewer 1 Report

The manuscript deals with a current and relevant theme for the area knowledge, especially for involving the nationally representative sample. The references used are current and pertinent to the theme of study. The objectives are clearly defined. The methods used are appropriate. However, it shows the need for discrete adjustments:

(a) Specifically regarding at-risk alcohol consumption, is necessary to define whether the doses of alcohol are consumed daily, weekly or monthly (line 80); and

(b) Table 3 (lines 135-136) is necessary to present the confidence intervals (95% CI) equivalent to the prevalence at baseline of lifestyle risk factors.

Also, I do not understand why only associations between lifestyle risk factors and CVD mortality were considered. Otherwise, why the associations between lifestyle risk factors and mortality from other diseases (for example, diabetes mellitus, gastrointestinal diseases, arthritis, cancer, lung disease) were not identified?

Author Response

In Responses to the Comments and Suggestions by Reviewer #1

Thanks for giving us an opportunity to revise and improve the quality of the manuscript for publication at IJERPH. The reviewers’ comments and suggestions are addressed point-by-point and highlighted in yellow color. Five references (#13, #30, #33, #34, and #35) are newly added in order to address the comments/suggestions.

(Reviewer #1)

Q1) Specifically regarding at-risk alcohol consumption, is necessary to define whether the doses of alcohol are consumed daily, weekly or monthly (line 80); and

ANS#1) Thanks for the comment. The doses of at-risk alcohol consumption is corrected as follows;

> 5 drinks per week for both men and women [13].

Q2) Table 3 (lines 135-136) is necessary to present the confidence intervals (95% CI) equivalent to the prevalence at baseline of lifestyle risk factors.

ANS#2) Thanks for the comment. 95% CI are also provided in Table 3 and stated as follows;

As illustrated in Table 3, the prevalence of lifestyle risk factors was 29.1% (95% CI, 28.2-30.0%) for smoking, 5.9% for at-risk alcohol consumption (95% CI, 5.5-6.4%), 26.3% (95% CI, 25.5-27.2%) for underweight/obesity, 71.3% (95% CI, 70.4-72.2%) for physical inactivity, and 10.3% (95% CI, 9.8-11.0%) for UWL.

Q3) Also, I do not understand why only associations between lifestyle risk factors and CVD mortality were considered. Otherwise, why the associations between lifestyle risk factors and mortality from other diseases (for example, diabetes mellitus, gastrointestinal diseases, arthritis, cancer, lung disease) were not identified?

ANS#3) Thanks for the comment. Unfortunately, the cause of deaths was reported to CVD or non-CVD due to not enough cases to be analyzed statistically. The association between risk factors and non-CVD mortality is now added to Tables 3-4. The following description regarding the relative risks of 4-risk factors and 5-risk factors for non-CVD mortality are also added;

Finally, the crude HR for non-CVD mortality was 1.721 (95% CI, 1.390-2.131) for one risk factor, 2.088 (95% CI, 1.678-2.599) for two risk factors, and 2.490 (95% CI, 1.916-3.236) for three or more risk factors, as compared to zero risk factor (HR=1). The HR for three or more risk factors remained significant (P=0.006) even after adjustments for all the covariates, while the HRs for one and two risk factors were no longer significant (P=0.207 and P=0.096, respectively) after adjustments for all the covariates.

Finally, the crude HR for non-CVD mortality was 1.722 (95% CI, 1.364-2.174) for one risk factor, 2.250 (95% CI, 1.782-2.842) for two risk factors, and 3.141 (95% CI, 2.433-4.055) for three or more risk factors. The HRs for two and three or more risk factors remained significant (P=0.047 and P<0.001, respectively) even after adjustments for all the covariates, while the HR for one risk factor was no longer significant (P=0.115) after adjustments for all the covariates.

Reviewer 2 Report

The paper is very interesting; however, the following concerns shoud be addressed:

The manuscript reads a bit dense with disjointed information and would benefit from re-organization of paragraphs; also, the Authors should better summarize some sections.

Please carefully revise wording and abbreviation throughout the manuscript.

The key role of diet and inflammation in the determination of cardiovascular risk (Gambardella et al. Atherosclerosis. 2016;253:258-261; doi: 10.1016/j.atherosclerosis.2016.08.041) should be better addressed.

The functional role of the adrenergic system in linking physical activity and cardiovascular health (Iaccarino G et al. Front Physiol. 2013 Aug 12;4:209) should be discussed in detail.

Author Response

In Responses to the Comments and Suggestions by Reviewer #2

Thanks for giving us an opportunity to revise and improve the quality of the manuscript for publication at IJERPH. The reviewers’ comments and suggestions are addressed point-by-point and highlighted in yellow color. Five references (#13, #30, #33, #34, and #35) are newly added in order to address the comments/suggestions.

 (Reviewer #2)

Q1) The manuscript reads a bit dense with disjointed information and would benefit from re-organization of paragraphs; also, the Authors should better summarize some sections.

ANS#1) Thanks for the comment. Introduction is revised to better address the rationality of the study;

“Participants in the Korean longitudinal study of aging (KLoSA) have his/her unique lifestyle pattern characterized by higher rates of smoking, heavy alcohol intake, and physical inactivity [9]. Despite the high prevalence of lifestyle risk factors in Korea [10], however, only one population-based study has been conducted to examine the association between lifestyle risk factors and mortality, reporting a significant relationship of unhealthy behaviors with an increased risk of all-cause, cancer, and non-cancer mortality in Korean adults [11]. More studies at a population level are necessary to provide more useful and practical information for policy making and diseases prevention in Korea.

Furthermore, the association between lifestyle risk factors and premature deaths from all causes and CVD may be modulated by additional covariates, including ethnicity, culture, religion, employment status, region, and others. Thus, exploring the associations between lifestyle risk factors and all-cause and CVD mortality in the KLoSA population would be of great significance for men and women in other cities of Korea and for men and women in other Asian countries with similar culture and lifestyle. In this population-based study, therefore, we investigated the association between lifestyle risk factors and all-cause and CVD mortality in Korean adults.”

Q2)  Please carefully revise wording and abbreviation throughout the manuscript.

ANS#2) Thanks for the comment. Wording and abbreviations have been checked and revised (and highlighted in yellow color), if necessary, throughout the manuscript.

Q3)  The key role of diet and inflammation in the determination of cardiovascular risk (Gambardella et al. Atherosclerosis. 2016;253:258-261; doi: 10.1016/j.atherosclerosis.2016.08.041) should be better addressed.

ANS#3) Thanks. In response to the comment, not considering diet is stated as a major study limitation;

Second, the current study did not include other significant risk factors, such as poor nutrition, prolonged sitting and sleeping, which may affect health conditions and mortality. In particular, poor quality of diets (e.g., high-carbohydrate or high-fat) may contribute to an increased risk of CVD morbidity [32] and mortality [33] by triggering oxidative stress and thereby inflammatory responses [34]. Unfortunately, however, the KloSA data failed to include dietary information from the participants. Future studies should include those risk factors to delineate the complexity of the association between lifestyle risk factors and mortality in a better way.

Q4) The functional role of the adrenergic system in linking physical activity and cardiovascular health (Iaccarino G et al. Front Physiol. 2013 Aug 12;4:209) should be discussed in detail.

ANS#4) Like in young adults, it is well known that older adults may have numerous health benefits from participating in physical activity of any intensity, including improvements of cardiovascular, metabolic, and immune system, collectively contributing the ability to maintain homeostasis. In particular, regular physical activity may prevent and/or ameliorate aging-associated decline in the β adrenergic responsiveness indicative of the overall function of the cardiovascular system, contributing to the clinical improvement in the cardiovascular system and decreased risks of CVD morbidity and mortality in elderly persons [29].

Reviewer 3 Report

This study examined the association between lifestyle risk factors and all-cause/CVD mortality using KLoSA dataset. The study presents results based on survival analysis and Cox proportional hazards models. My major concerns the rationale of introduction is very weak, and the findings are poorly discussed.

First, the identified risk factors have been widely studied in previous literature. It is not clear what the research gap is and what the authors expect to add in the scientific literature.

Second, in addition to lifestyle factors, environmental and occupational factors are also significant predictors of all-cause/CVD mortality. However, it is not clear how and why the authors exclude these factors. They are not adequately explored and explained in this study.

Third, the 2nd to 7th paragraphs of Discussion section are more like a literature review, rather than a discussion. The lack of discussion on how this study fills the research gap and contribute new findings downplays the value of this article. It is difficult to conclude anything meaningful based on current information.

Since the authors have valuable data, I suggest the authors consider the above suggestions to improve the robustness of this study.

Author Response

In Responses to the Comments and Suggestions by Reviewer #3

Thanks for giving us an opportunity to revise and improve the quality of the manuscript for publication at IJERPH. The reviewers’ comments and suggestions are addressed point-by-point and highlighted in yellow color. Five references (#13, #30, #33, #34, and #35) are newly added in order to address the comments/suggestions.

(Reviewer #3)

My major concerns the rationale of introduction is very weak, and the findings are poorly discussed.

Q1) First, the identified risk factors have been widely studied in previous literature. It is not clear what the research gap is and what the authors expect to add in the scientific literature.

ANS#1) Thanks for the comment. To the best of our knowledge, only one population-based study examined the association between lifestyle risk factors and all-cause and cancer-mortality in Korea. Therefore, we believe that more studies are necessary to explore the association between lifestyle risk factors and mortality, which would be more informative for policy making and diseases prevention. In response to the comment, Introduction is revised to better address the necessity of the current study.

“Participants in the Korean longitudinal study of aging (KLoSA) have his/her unique lifestyle pattern characterized by higher rates of smoking, heavy alcohol intake, and physical inactivity [9]. Despite the high prevalence of lifestyle risk factors in Korea [10], however, only one population-based study has been conducted to examine the association between lifestyle risk factors and mortality, reporting a significant relationship of unhealthy behaviors with an increased risk of all-cause, cancer, and non-cancer mortality in Korean adults [11]. More studies at a population level are necessary to provide more useful and practical information for policy making and diseases prevention in Korea.

Furthermore, the association between lifestyle risk factors and premature deaths from all causes and CVD may be modulated by additional covariates, including ethnicity, culture, religion, employment status, region, and others. Thus, exploring the associations between lifestyle risk factors and all-cause and CVD mortality in the KLoSA population would be of great significance for men and women in other cities of Korea and for men and women in other Asian countries with similar culture and lifestyle. In this population-based study, therefore, we investigated the association between lifestyle risk factors and all-cause and CVD mortality in Korean adults.”

Q2) Second, in addition to lifestyle factors, environmental and occupational factors are also significant predictors of all-cause/CVD mortality. However, it is not clear how and why the authors exclude these factors. They are not adequately explored and explained in this study.

ANS#2) Thanks for the comment. In response to the critics, additional covariates, including employment status (currently employed or not employed), religion (no religion or protestant, catholic, Buddhist, and others), type of housing (apartment or general house), and region (urban or rural), are now added, and they were also adjusted for the calculations of relative risks for all- and cause-specific mortality of lifestyle risk factors.

Q3) Third, the 2nd to 7th paragraphs of Discussion section are more like a literature review, rather than a discussion. The lack of discussion on how this study fills the research gap and contribute new findings downplays the value of this article. It is difficult to conclude anything meaningful based on current information.

ANS#3) Thanks. In response to the comment, the 2nd – 7th paragraphs of Discussion are now revised as follows;

“The current findings support and extend those of previous studies reporting the associations between health behaviors and premature death from all and specific causes in Western populations; the healthier behaviors the lower mortality risk or vice versa. By analyzing data from a healthy ageing: a longitudinal study in Europe (HALE) study, for example, Knoops et al. [13] showed that adherence to healthful lifestyle behaviors, including Mediterranean diet, physical activity, moderate alcohol consumption, and not smoking, was associated with a lower rate of all-cause and specific-cause mortality in European older adults. Using data obtained from a nurses’ health study (n = 78 865) and a health professionals follow-up study (n = 44 354), Li et al. [14] also examined five low-risk lifestyle factors (i.e., non-smoking, healthy body weights, physical activity, moderate alcohol intake, and a healthy diet) and mortality and found that compared with those who had zero-low risk factor (HR, 1), those who had 5 low-risk factors had significantly lower risks of all-cause mortality (HR, 0.26; 95% CI, 0.22-0.31), cancer mortality (HR, 0.35; 95% CI, 0.27-0.45), and CVD mortality (HR, 0.18; 95% CI, 0.12-0.26).

On the other hand, Krokstad et al. [15] examined the association between lifestyle risk factors - including smoking, alcohol consumption, diet, physical activity, sedentary behaviors, sleep, and social participation - and mortality in a Norwegian cohort study (n=36,911 adults aged 20~69 years) and found that all the risk factors, except for diet, were significantly associated with an increased risk of 14.1-year-all-cause and CVD mortality. By conducting a population-based cohort study (n= 231,048 Australians aged 45 years or older), Ding et al. [16] also examined the associations between six risk factors - smoking, alcohol use, dietary behavior, physical inactivity, sedentary behavior, and sleep - and mortality and found a positive linear trend of mortality risk according to an increasing number of risk factors. In particular, combinations of physical inactivity, prolonged sitting, and long sleep duration as well as combinations of smoking and high alcohol use were the most important determinants of all-cause mortality, implying individual as well as additive effects of those lifestyle risk factors on mortality. Together, those previous findings support the associations between lifestyle behaviors and the risk of premature death from all causes and diseases in Asian as well as Western populations.”

Round 2

Reviewer 2 Report

No further comments.

Reviewer 3 Report

The authors’ effort on revision is worth encouraging. However, the rationale of the introduction and the findings in the revision are remaining weak. In addition, English editing is needed.

Comment on ANS#1: The identified risk factors have been widely studied in previous literature “internationally.” It is not clear why the Korean population is different from other study population.

Comment on ANS#2: It is not clear how the newly added covariates (employment, religion, type of housing, and region) can be linked to environmental and occupational factors (e.g., exposure to air pollutants, living nearby industrial areas, and exposure to hazardous chemicals in the workplaces). Still, they are not adequately explored and explained in the revision.

Comment on ANS#3: The authors described how their results support or are similar to previous studies. Corresponding to Comment on ANS#1, it is not clear how the findings of this study can fill the research gap and what the authors added to the scientific literature. It is still difficult to conclude the originality/novelty/significance of the manuscript based on current information.